# A New Approach for Measuring Sentiment Orientation based on Multi-Dimensional Vector Space

## Abstract

This study implements a vector space model approach to measure the sentiment orientations of words. Two representative vectors for positive/negative polarity are constructed using high-dimensional vector space in both an unsupervised and a semi-supervised manner. A sentiment orientation value per word is determined by taking the difference between the cosine distances against the two reference vectors. These two conditions (unsupervised and semi-supervised) are compared against an existing unsupervised method (Turney, 2002). As a result of our experiment, we demonstrate that this novel approach significantly outperforms the previous unsupervised approach and is more practical and data efficient as well.

## 1 Introduction

Previous research in sentiment analysis or opinion mining mostly focus on supervised methods, which requires labeled training data to identify properties of unseen input, and then classify input later. Probabilistic methods in particular often calculate which class a word or phrase most likely bears and then make predictions regarding the label of a given target text, using those estimations. While such methods have widely been adopted, there are few examples which measure the likelihood of lexical items in an unsupervised or semi-supervised manner. However, there still exist situations where sentiment analysis should be performed using non-labeled datasets. In such cases, discovering information regarding the sentiment orientation of vocabulary in a non-supervised fashion becomes essential.

Our approach employs VSM (Vector Space Models) as its main component. VSM is deeply related to the distributional hypothesis (Turney and Pantel, 2010). The distributional hypothesis states that words in similar contexts tend to have similar meaning (Rubenstein and Goodenough, 1965; Schütze and Pederson, 1995; Deerwester et al., 1990). Traditionally, the relation of two words in a 'similar context' has been distinguished into two classes: syntagmatic or paradigmatic (Murphy, 2003; Sahlgren, 2006). Syntagmatic relations are concerned with whether or not two entities are in a co-occurrence relation, and paradigmatic relations are concerned with whether the two items in question are interchangeable (substitution relation). Many collocation models using N-grams and Point-wise Mutual Information (PMI) analyze the former-type of word relations. On the other hand, recent dense vector-based models (Skip-gram, Continuous Bag-of-Words) exploit the paradigmatic relation, and thus they both give a high weight to the similarity of words if they share similar neighboring entities (Mikolov et al., 2013).

Our work can best be understood as an exploration to find a sentiment dimension over a multi-dimensional vector space, which is constructed from the relations of words in a corpus. However, it is too complex to extract a specific type of relation between whole words on such a high-dimensional space. Thus, we start by selecting a small set of words that are believed to have an emotional value for a topic. We refer to these words as 'point words' and use them to construct a latent sentiment dimension. Our method for choosing these point words can be divided into two sub-types: unsupervised or semi-supervised.

If we use a supervised learning approach, one simple, intuitive way of calculating the sentiment

orientation of a word is to compute the log-likelihood ratio of probabilities in terms of occurrence per sentiment label. Should the labels not be given, one alternative method would be to use the similarity between words. Following the principle of the Distributional Hypothesis, we assume that positive or negative words will tend to share similar contexts relative to their opposing stance.

We argue that collocation-based methods are not a practical choice for obtaining the similarity due to data sparsity which is inherent to the model. Our implementation of PMI-IR method (Turney, 2002) for this problem demonstrates this point and hints as to why dense vector space models should be used instead. For this purpose, we also compared two well-known vector models (Word2Vec and GloVe), as the latter model mainly depends on the collocation relations of words, while Word2Vec's Skip-gram model is dependent on the paradigmatic relations between words.

In the unsupervised condition, a dimensionality reduction algorithm is implemented to search for sentiment dimension using the selected point words. Under the semi-supervised condition, we depend on external estimations of the terms in order to skip the exploration stage. We believe that this study can be a direct comparison with the previous work of (Turney, 2002), because we use the same seed information and a similar procedure for calculating the semantic orientations of the words.

## 2 Related Work

Although the majority of previous studies on sentiment analysis have preferred to use supervised methods, some researchers have tried to develop unsupervised or semi-unsupervised approaches. For instance, Turney (2002) suggests the PMI-IR algorithm to estimate the semantic orientation of a phrase for the unsupervised classification of various reviews. He uses two pre-chosen words ('poor' and 'excellent') to calculate the semantic orientation of the target phrases, which is defined as the relative PMI difference of the phrase from the two seed words. His work borrows heavily on the theory of semantic orientation of adjectives by Hatzivassiloglou and McKeown (1997). In this study, the authors discuss the existence of linguistic constraints on the semantic orientations of words in conjunctions. In line with the approach of Turney

(2002), Zagibalov and Carroll (2008) attempt to develop an automatic selection process for seed words in Chinese texts for unsupervised classifications.

One major constraint on Turney (2002) is the availability of a corpus to calculate the relevant PMI. If a given equipped corpus is not big enough for a PMI analysis, the problem of data sparseness will arise, and the PMI values become suspect. Note that Turney (2002) used a search engine (AltaVista) for his experiments, which contained 350 million web pages at the time. In our experiment, we instead use Yadex.com (`http://www.yandex.com/`), as it provides a more reliable Near operator among the current major search engines.[1] With the operator, words in a query have to be within 10 words of each other, regardless of order. We used the same formula (Eq. 1) from Turney (2002) for our experiment:

$$SO(prhase) =$$
$$log_2 \left[ \frac{hits(phrase, NEAR \text{ "excellent"}) \; hits(\text{"poor"})}{hits(phrase, NEAR \text{ "poor"}) \; hits(\text{"excellent"})} \right] \quad (1)$$

Note that hits(query) is the number of the returns, given the query. Additionally, we add 0.01 to hits when the number of the hits is zero, in order to prevent division-by-zero.

In contrast to collocation models, some researchers attempt to apply neural probabilistic language models to measure the semantic similarities of words based on context-window methods (Mikolov et al., 2013; Collobert and Weston, 2008), and word embedding methods (e.g., Word2Vec) have been found more effective for various tasks in NLP than other traditional techniques (Baroni et al., 2014). Continuous Bag-of-Words (CBOW) models and Skip-gram models used in Mikolov et al., (2013) both place a high weight on the similarity of words if they share similar neighboring entities.

Additionally, we consider another word-embedding model (GloVe). Unlike Skip-gram or the CBOW architecture of Word2Vec, GloVe uses the ratios of words' probability of co-occurrence to learn word vectors (Pennington et al., 2014). We note that GloVe is a kind of word embedding model, but its vector space is constructed differently from Word2Vec, because it adopts the collocation modeling of words.

---

[1] Yandex reported that it indexes more than 4 billion pages written in the Latin Alphabet with the majority of them being in English (`https://yandex.com/company`

`/press_center/press_releases/2010/2010-05-19`)

We note that the implication of the Skip-gram/CBOW architecture is very similar to the concept of paradigmatic relation of words. This is due to the fact that the distance between any two words in a paradigmatic relation is minimized when they share the most similar neighbors. For example, in a minuscule corpus which bears only two sentences ("This movie is very good" and "This movie is very bad"), the two words ('good' and 'bad') will likely have a high cosine similarity in the Word2Vec model.

This aspect can cause unexpected results when such a model is employed for clustering a set of items that share similar emotions, because two words in paradigmatic relations often instantiate a contrastive relation (e.g., antonym). However, as noted in Mikolov et al. (2013), words seem to have multiple syntactic/semantic relations to each other, and the Word2Vec model helps to observe the multiple degrees of similarity for words. From this perspective, we might find a specific relation to them in a subspace of the original vector space if the necessary vector calculation operations are known.

A feature-space approach (Osgood, 1952; Smith and Medin, 1981; Waltz and Pollack, 1985) discusses how words are represented by feature vectors whose attributional factors are annotated by human participants. In a similar vein, the Conceptual Space theory of Gärdenfors (2000) provides a theoretical framework for understanding the multiple degrees of similarity in the space. He also suggests using dimensionality reduction algorithms (e.g., Multidimensional scaling) to explore the quality dimensions of multi-dimensional vector space and claims that applying these algorithms to the similarity-based vector space will generate an ordering relation for data points on an interested domain.

Another relevant prior work is the Word-Space model, which is a vector-based computational model for the semantic similarity of words (Schütze, 1993; Sahlgren, 2006). The Word-Space model, as the name implies, models word meaning with a spatial representation. Thus, semantic similarity is represented as proximity in n-dimensional space.

Our goal is to find or construct a sentiment dimension from the similarity-based vector space of words. The vector representation for the sentiment dimension will be our 'interested domain' and the 'ordering relation' on the domain will map each word object to a real valued point, indicating the level of its 'positive' or 'negative' significance.

## 3 Methods and Data

The purpose of our experiment is to find the sentiment orientations of words in a corpus and evaluate the effectiveness of the information by conducting an unsupervised/semi-supervised classification on our movie review datasets.

### 3.1 Data

Our data consists of two movie review corpora: one of which is the IMDB movie dataset (Maas et al., 2011) and the other is the Stanford Sentiment Treebank (Socher et al., 2013). The IMDB dataset provides 25,000 movie reviews for training and 25,000 for testing. The corpus also contains the expected polarity values for all individual tokens occurring in the reviews. This dataset is used for our vector space construction and the polarity values per word are employed under the semi-supervised condition.

The Stanford Sentiment Treebank is a corpus based on the 11,855 movie reviews presented by Pang and Lee (2005), and all 215,154 phrases are manually annotated by three judges per phrase. Because we want the classification task to be binary, reviews with neutral labels are excluded from the dataset, and the number of reviews in the resulting dataset is 9,613. We select this corpus as our test dataset since it allows us to compare our sentiment orientation values with the annotated polarity value for each word.

### 3.2 Experiment Methods

The first step of our methodology is to obtain a set of point words which are assumed to express positive/negative sentiment. Using the pattern extrac-

| | First Word | Second Word | Third Word |
|---|---|---|---|
| 1 | JJ | NN or NNS | anything |
| 2 | RB, RBR or RBS | JJ | not NN nor NNS |
| 3 | JJ | JJ | not NN nor NNS |
| 4 | NN or NNS | VB, VBD | not NN nor NNS |
| 5 | RB, RBR, or RBS | VBN, or VBG | anything |

Table 1. Pattern rules of tags for extraction two-word phrases (third word excluded)

tion rules (Table. 1) from Turney (2002), we obtained 17,716 phrases by applying these rules to the Stanford Sentiment Treebank.

We use the adjectives or adverbs of the phrases for our point words, since these syntactic categories have been found very useful for recognizing subjectivity in written texts (Hatzivassiloglou and Wiebe, 2000; Wiebe et al., 1999; Bruce and Wiebe, 2000).

To validate this assumption, we POS-tagged the word tokens of the Stanford corpus and observe the variance of sentiment ratings of words depending on their POS-tag (Fig. 1).

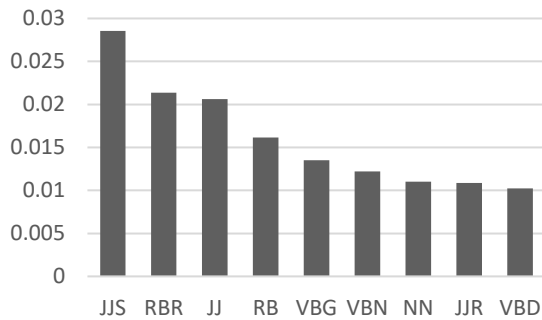

Figure 1. Variances of polarity values per POS-tags

Not surprisingly, the main tags of adjective and adverb occupy the top places in the ranking, indicating that terms bearing these tags are often used with stronger subjectivity. Numerically, the top four tags (JJS, RBR, JJ, and RB) make up 37% of total variances over the whole dataset.

Based on the theory and the observation, we select the top K modifiers (adjectives or adverbs) from the extracted patterns. The number of the modifiers are automatically determined by choosing a minimum frequency for the extracted phrases.

### 3.2.1 Unsupervised methodology

After the point words (K modifiers) are obtained, a high-dimensional vector space is implemented to build a point-wise distance matrix for the selected words, resulting in a K×K matrix. Since the distances between the tokens are measured by Cosine-distance, the distance data is Euclidean. As mentioned in Section 3.1, we use the IMDB dataset for vector space construction.

A local structure-oriented dimensionality reduction algorithm, the Principal Component Analysis (PCA) is then used to find the sentiment dimension between the modifiers. In the analysis, we only use

the value of two for the number of dimensions to project the entities into the reduced space.

Based on the principle of the Distribution Hypothesis, we assume that semantically close modifiers are closer to each other than the opposites. Thus, the principal component of the PCA for the emotional terms will represent the positive/negative aspects of their collective meaning.

When the dimensionality reduction phase is completed, it is possible to observe correlations between values on the found dimension and the gold-standard dataset (the annotated values of Stanford Sentiment Treebank). Since the signs of the coefficients are irrelevant for our purposes, only absolutes are considered.

Now, we can determine the two sets of words distinguished by the origin of zero on the principal axis. To construct two reference vectors, simple vector averaging is used, and the vectors are classified as positive or negative by the criterion of being closer to the vector of the seed word ("excellent"). Note that this method was inspired by Turney (2002) and makes our study comparable to the previous work. We define the sentiment orientation of a word using Equation (2):

$$SO(w) = CosDst(Vec_{pos}, w) - CosDst(Vec_{neg}, w) \quad (2)$$

CosDst means Cosine distance in the equation. If the mean of the sentiment orientations in a review post is less than zero, the review is labeled as 'negative', and is 'positive' otherwise. We note that this approach allows us to calculate the orientations of all words in the vocabulary, unlike Turney's phrase-oriented approach.

### 3.2.2 Semi-supervised methodology

This method does not use the dimensionality reduction algorithm to distinguish the point words into two sets, but instead employs the expected star ratings of the tokens from the IMDB dataset. The ratings represent how strongly a word belongs to positive or negative sentiment polarity.

Thus, in this case, we start with almost certain information on the polarity of the words. Because adjectives and adverbs are generally used with their own static stance, we believe that the information can be applied to unseen texts over different domains. Since this semi-supervised setting is designed to compare with the unsupervised condition, the remainder of the experiment methodology is identical to the unsupervised methodology.

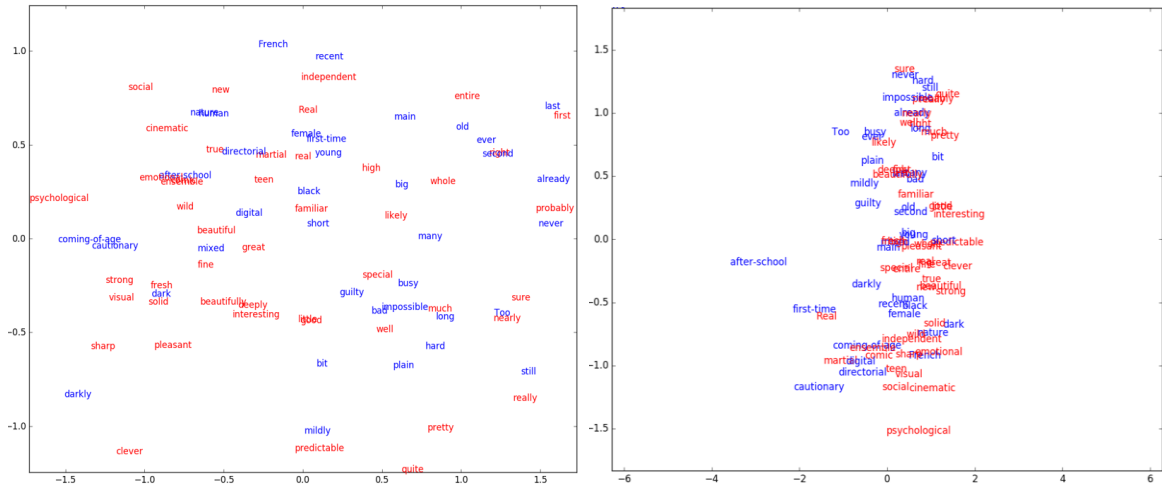

Figure 2. 2-D embedding of the Principal Component Analysis of the point words. The left figure shows the result from Word2Vec-based space and the right figure from GloVe model. The color is coded by the ratings of Stanford Sentiment Treebank (red for positive and blue for negative).

### 3.2.3 PMI-IR methodology

We replicate the PMI-IR algorithm (Turney, 2002) against the Stanford corpus. The queries on the 17 thousand phrases are sent to the Search API of Yandex.com (https://yandex.com/search/) and the hits of the phrases with the two seeds ("excellent" and "poor") are disregarded if both of the two hits are less than four counts.

Since the method is only applicable to a review that contains at least the one of the defined pat terns (Table. 1), the test dataset consists of 7,646 posts out of the 9,613 reviews and the average number of phrases per review is 2.25.

## 4 Results

Our first observation in the unsupervised setting is the visualization of the PCA result on the selected

| Model | Pearson | Spearman |
|---|---|---|
| Word2Vec (size:3) | 0.33 | 0.32 |
| Word2Vec (size:5) | 0.29 | 0.29 |
| Word2Vec (size:7) | 0.27 | 0.26 |
| Word2Vec(size:10) | 0.25 | 0.24 |
| GloVe (size:3) | 0.28 | 0.33 |
| GloVe (size:5) | 0.27 | 0.30 |
| GloVe (size: 7) | 0.28 | 0.34 |
| GloVe (size: 10) | 0.28 | 0.34 |

Table 2. Correlations between the unsupervised ratings and annotated ratings

modifiers using Word2Vec (Skip-gram, no_components: 100) and GloVe models (epochs: 30, no_components: 100). Fig. 1 shows exemplary results of the 82 tokens, which are from the extracted patterns of the Stanford dataset. The point words are colored red (positive) or blue (negative) by the middle value on the annotation scale.

The principal dimension in the embedding by PCA gives us an ordering relation between the terms. As explained in Section 3, we interpret the dimension as a sentiment domain in the high-dimensional vector space of words. The correlation coefficients in Table 2 demonstrates how the dimension correlates with our gold-standard dataset (the signs of the correlations are ignored). Although there undoubtedly exists noise in the results, we could still find correlations between the unsupervised ratings and the annotated values. We also note that varying the size of context window for the embedding models produces slightly different patterns. As Table 2 shows, the Word2Vec model tends to lose its potency as the size of context window increases, while GloVe remains the same.

Our semi-supervised approach does not use the procedure incorporating the PCA, but the vector averaging and classification processes are identical to the unsupervised condition.

Fig. 2 represents the accuracy results of the two settings for the 9,613 reviews, while changing the minimum frequency of the extracted patterns in which the modifiers occurred. The cutoff frequency ranges from 2 to 9 and the corresponding number of the word tokens varies from 495 to 30. We set the context window size to 3 for the

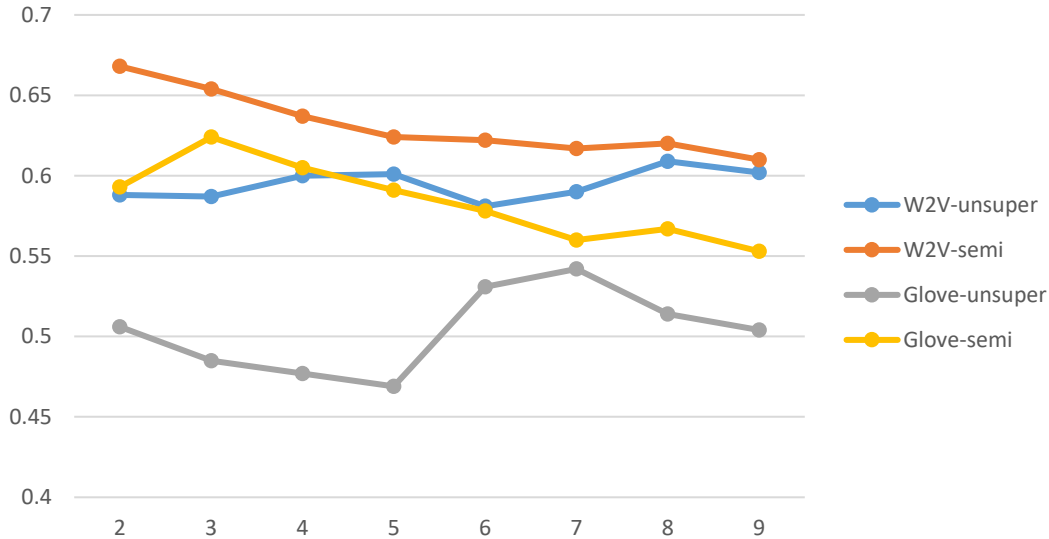

Figure 2. Accuracy of Classifications. The x-axis indicates the cutoff frequency.

Word2Vec (Skip-gram based) and 10 for the GloVe model.

As can be seen in the graph, the Word2Vec conditions often outperform the GloVe conditions, and the unsupervised Word2Vec generally records higher accuracy than the GloVe. The highest accuracy is achieved by semi-supervised Word2Vec (66%) and the lowest score is falls to the unsupervised GloVe model (lower than chance). The best result for unsupervised Word2Vec is 61% (cutoff: 8), and the variance of the accuracies is relatively small compared to other models. Decreasing the number of the point words worsens the performance of all the semi-supervised models.

The general pattern of Fig. 2 is found in the results of our experiment combined with the PMI-IR algorithm against the reduced test dataset (Table 3). Semi-supervised/unsupervised Word2Vec models generally record higher accuracies than the GloVe models. On top of this, the semi-supervised methods show a better performance than the unsupervised settings.

| Model | Cutoff | Accuracy |
|---|---|---|
| W2V (semi-supervised) | 8 | 0.66 |
| W2V (unsupervised) | 5 | 0.63 |
| GloVe (semi-supervised) | 3 | 0.63 |
| GloVe (unsupervised) | 9 | 0.57 |
| PMI-IR | N/A | 0.57 |

Table 3. highest accuracy of classification results of the selected test set (7,646 reviews)

The PMI-IR produces the lowest accuracy along with the unsupervised GloVe model (57%). As mentioned in Section 3, the PMI-based approach is based on the returned search results from Yandex engine, while our methods use vector space from the IMDB corpus of 50,000 movie reviews. The classification by PMI-IR is processed by calculating the mean of semantic orientations from the extracted phrases per review as suggested in Turney (2002).

## 5  Discussion

We present an unsupervised/semi-supervised approach that measures the sentiment orientation of words using vector space models. The core idea of our approach is to find a sentiment dimension from a high-dimensional vector space of words and use the extracted information to calculate the polarity of an individual word. We attempted to see whether or not the obtained sentiment orientations are useful by the sentiment classification task of movie reviews.

Our approach borrows from the general paradigm of Word Space models (Schütze, 1993) and is inspired by the theoretical tools of feature space models (Osgood, 1952; Smith and Medin, 1981; Waltz and Pollack, 1985; Gärdenfors, 2000). The general framework of our experiment is based on Turney (2002) and our computational methods rely on recent successes in word-embedding research (Mikolov et al., 2013; Pennington et al., 2014).

Our unsupervised methodology (Word2Vec-based) outperforms the PMI-IR approach, which is

a well-known unsupervised tool in a sentiment classification task. Additionally, the unsupervised Word2Vec setting records much higher accuracy than the unsupervised GloVe model. We suspect that the cause of this low performance is from the common feature that the both models are based on word collocations.

Considered that the PMI-IR approach employs the results of a major search engine (Yandex.com), the low accuracy indicates that data-sparsity is a very difficult issue for a collocation-based method to overcome. Even the word-embedding model (GloVe) seems unable to break free of this problem, as the models show lower results than the those using Skip-gram. This conjecture is supported by the observation that the GloVe model does not lose its correlation coefficients as the context size increases. We note that Skip-grams essentially exploit paradigmatic relations between words and produce denser vector spaces for the relation of words. Thus, methods based on dense vector modeling should be more robust for the data-sparsity problem in our task, as demonstrated by our experiment.

It is worth noting that PMI-IR suffers from data-deficiency. In many practical situations, it is hard to collect information on the collocations of phrases using a sufficient search engine service, as major search engines often constrain the queries of users by their policies or do not provide the 'Near' operator for queries.

One issue in our study is how to find **optimal** reference vectors to represent the sentiment polarity of a vector space. We tried to approximate by using a traditional operation (vector averaging). Note that the performance of the semi-supervised approach did not dramatically increase with the added number of the word tokens. This likely indicates that the type of vector calculation is not efficient for the purpose in this research. Additionally, the results of our experiment do not meet the high standards of a supervised approach (usually above 80%). However, we believe that there is potential for huge improvement, as the reference vectors can be constructed in an optimal way to represent the sentiment domain between words. We predict that there could be two possible ways to achieve this goal. The first way is to select the point words that best capture the landscape of the sentiment entities in a corpus. And the second way is to exploit more relevant vector space models than the embedding methods used in this study. We leave such explorations for future work.

## 6 Conclusion

In this study, we introduce a novel approach which implements distributional semantic models to measure the sentiment orientation of a word. We divide our experiment into two separate types (unsupervised or semi-supervised) and compare the results with a previous unsupervised approach (PMI-IR). Our new unsupervised methodology (Word2Vec-based) outperforms the existing approach, using much smaller datasets, while being robust enough to overcome the problem of data-sparseness.

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
