# Peer review of "A New Approach for Measuring Sentiment Orientation based on Multi-Dimensional Vector Space"

_ACL 2017 — decision unknown_

[Official Review · Reviewer 1 · rating 1 · confidence 2]
soundness 3 · originality 3 · clarity 3 · impact 3 · substance 1 · appropriateness 5 · meaningful comparison 3 · presentation format Poster

# Summary

This paper presents an empirical study to identify a latent dimension of
sentiment in word embeddings.

# Strengths

 S1) Tackles a challenging problem of unsupervised sentiment analysis.

 S2) Figure 2, in particular, is a nice visualisation.

# Weaknesses

 W1) The experiments, in particular, are very thin. I would recommend also
measuring F1 performance and expanding the number of techniques compared.

 W2) The methodology description needs more organisation and elaboration. The
ideas tested are itemised, but insufficiently justified. 

 W3) The results are quite weak in terms of the reported accuracy and depth of
analysis. Perhaps this work needs more development, particularly with
validating the central assumption that the Distributional Hypothesis implies
that opposite words, although semantically similar, are separated well in the
vector space?

[Official Review · Reviewer 2 · rating 2 · confidence 5]
soundness 3 · originality 3 · clarity 4 · impact 3 · substance 2 · appropriateness 5 · meaningful comparison 3 · presentation format Poster

- Strengths
This paper deals with the issue of finding word polarity orientation in an
unsupervised manner, using word embeddings.

- Weaknesses
The paper presents an interesting and useful idea, however, at this moment, it
is not applied to any test case. The ideas on which it is based are explained
in an "intuitive" manner and not thoroughly justified. 

- General Discussion
This is definitely interesting work. The paper would benefit from more
experiments being carried out, comparison with other methods (for example, the
use of the Normalized Google Distance by authors such as (Balahur and Montoyo,
2008) - http://ieeexplore.ieee.org/abstract/document/4906796/) and the
application of the knowledge obtained to a real sentiment analysis scenario. At
this point, the work, although promising, is in its initial phase.